# Oxidative Stress and Autophagy Mediate Anti-Cancer Properties of Cannabis Derivatives in Human Oral Cancer Cells

**DOI:** 10.3390/cancers14194924

**Published:** 2022-10-08

**Authors:** Lionel Loubaki, Mahmoud Rouabhia, Mohamed Al Zahrani, Abdullah Al Amri, Abdelhabib Semlali

**Affiliations:** 1Héma-Québec, 1070, Avenue des Sciences-de-la-Vie, Québec, QC G1V 5C3, Canada; 2Groupe de Recherche en Écologie Buccale, Faculté de Médecine Dentaire, Université Laval, Québec, QC G1V 0A6, Canada; 3Department of Biology, College of Science, Imam Mohammad Ibn Saud Islamic University (IMSIU), Riyadh 11623, Saudi Arabia; 4Biochemistry Department, College of Science, King Saud University, P.O. Box 2455, Riyadh 11451, Saudi Arabia

**Keywords:** cannabinoid mixture, oral cancer, apoptosis, autophagy, oxidative stress, MAPK, STAT and NF-κB pathways

## Abstract

**Simple Summary:**

The legalization of cannabis in 2018 in Canada has generated a heated public debate. The subject is complex, controversial, and completely opposite currents of thought clash mainly regarding its recreational use. The therapeutic efficacy of cannabis is very limited and still needs to be confirmed or refuted. However, our recent work has shown that at low doses, cannabinoids (Δ9-THC and Δ8-THC), which are the main constituents of cannabis, are beneficial against oral cancer. In this current study, we showed that a mixture of cannabinoids (CM) can induce oral toxicity in cells by damaging the DNA and activating the mechanisms of autophagy and apoptosis along with inhibiting many cancer progression pathways such as MAPKase, STATs and NF-κB pathways. These data demonstrated clearly the potential beneficial effect of CM at low concentrations for oral cancer therapy.

**Abstract:**

Cannabinoids, the active components of cannabis exert palliative effects in cancer patients by preventing nausea, vomiting and pain as well as by stimulating appetite. Recent studies indicated that cannabinoids could be helpful in treating certain rare forms of cancer and other inflammatory diseases. The objective of this study was to investigate the cytotoxic effect of a cannabinoid mixture (CM) in oral cells. Thus, normal and cancer gingival cells were treated with different concentrations of CM to evaluate their proliferation by MTT assay, cytotoxicity by using LDH assay, colony formation with crystal violet and migration by the scratch method. In addition, apoptosis, autophagy, oxidative stress, antioxidant level, DNA damage and the mitochondrial membrane potential (ΔΨm) generated by proton pumps were measured by flow cytometry. Furthermore, deactivation of the key signaling pathways involved in cancer progression such as NF-κB, ERK1/2, p38, STAT1, STAT3, STAT5 was also evaluated by this technique. These outcomes indicate that CM, at a concentration higher than 0.1 µg/mL, provokes high cytotoxicity in Ca9-22 oral cancer cells but not in GMSM-K gingival normal cells. Apoptosis, autophagy, antioxidant levels and mitochondrial stress as well as DNA damage in oral cells were increased following exposure to low concentration (1 µg/mL). In addition, major signaling pathways that are involved such as MAPKase, STATs and NF-κB pathways were inhibited by CM as well as cell migration. Our results suggest that cannabinoids could potentially have a beneficial effect on oral cancer therapy.

## 1. Introduction

Oral cancer (OC) is one of the most important contributors to mortality worldwide [1,2]. It is a deadly disease that continues to disrupt millions of people’s lives around the world [1]. It is now considered the sixth leading cause of morbidity and mortality globally. According to the American Cancer Society, more than 54,000 new cases and 11,230 deaths were reported in 2022 [3]. Oral cancer is a complex and multifactorial disease associated with different risk factors that can be modified such as alcohol consumption, tobacco use, a low fiber diet [4] and HPV infection with a 30-fold increased risk for individuals who both smoke and drink heavily. Furthermore, oral dysbiosis has been linked to periodontal disease and was reported to be a possible etiologic factor for oral squamous cell carcinoma, either isolated or synergistic with other agents [5]. It was reported that OC has been associated with increased oxidative stress, as lipid peroxidation, and reduced antioxidants were reported in patients suffering from stage II, III and IV oral cancer [6] as well as autophagy [7,8]. Indeed, the involvement of reactive oxygen species (ROS) in cancer development has been studied for decades and there is sufficient evidence that implicates them in the multistage theory of carcinogenesis as they are proposed to cause diverse DNA alterations such as: punctual mutations, DNA base oxidation, strand breaks, mutation of tumor suppressor genes and can induce overexpression of proto-oncogenes [9,10]. Moreover, an increase in ROS levels has been linked to tissue injury and/or damage to intracellular components and are involved in a wide range of crucial physiological processes such as cell cycle progression, antiapoptotic mechanisms, invasion, metastasis and angiogenesis, which contribute to different types of cancer pathogenesis [9,11,12,13,14]. Furthermore, autophagy, a process acting like a “double-edged sword” by which cells degrade old and defective cellular components, has been proposed as an important mechanism of cell death (Jung et al., 2020; Patil et al., 2015). Thus, both ROS and autophagy are key potential therapeutic targets for OC to be treated. The disease-conventional treatment is based on surgery, radiotherapy, chemotherapy or a combination. These therapies present side effects that have a huge impact on the quality of life of patients [15]. Despite multiple drug combinations and regimens, all subjects with advanced OC, similarly to other solid tumors, inevitably develop conventional treatment resistance [16]. Recently, natural products have emerged to overcome antibiotic resistance in cancer. In addition, several clinical works have reported that natural products such as cannabis or their derivatives have safe potent anti-cancer properties without side effects, and therefore can be an alternative solution to current conventional treatments well-known for their adverse reactions and for being non-selective. However, cannabis legalization in Canada back in 2018 sparked heated public debate and has worried since then some public health experts. The subject is complex and controversial and there are completely opposite currents of thoughts, a clash mainly when it comes to its recreational use. Cannabis contains bioactive molecules known for their beneficial effects. However, a lot of reports show that certain cannabinoids, key bioactive components of cannabis playing an important role as an aid in controlling symptoms and managing side effects, and many cannabinoid-containing medicines have been developed to prevent pain, nausea and vomiting [17]. Cannabis contains numerous molecules, including more than 60 chemical compounds called cannabinoids, of which the two main ones are ∆9-tetrahydrocannabinol (THC) and cannabidiol (CBD) [18]. They exert their effects through the endocannabinoid receptors (CB1 and CB2) and can modulate multiple cancer-related pathways such as MAPKase [19,20], NF-κB [21,22], Wnt/beta-catenin [23], mTOR [24,25] and PPAR-γ [26,27], some of which play an important role in ROS and autophagy signaling. In addition, we have recently shown that inhibition of these pathways by Δ(9)-THC and Δ(8)-THC resulted in suppressed cancer cell growth, cell cycle progression and selectively promoted apoptosis and autophagy in oral cancer cells [28]. The main objective of this study was to investigate the effect of a cannabinoid mixture (CM) on oral toxicity and focuses on the different mechanisms by which CM exerts its potential anti-oral-cancer properties.

## 2. Methods and Materials

### 2.1. Cell Culture

Ca9-22 cells (human gingival carcinoma cell line) were cultured in RPMI-1640 (RPMI-1640; Corning, Manassas, VA, USA). GMSM-K cells (human polyclonal oral epithelial cell line) were provided by Dr. Grenier (Université Laval). The GMSM-K cell line was constructed by Gilchrist et al. (2000) who transfected oral epithelial cells with the shuttle vector plasmid pZ189 containing the T antigen coding region and the simian virus 40 (SV40) origin of replication. GMSM-K cells were in Dulbecco’s Modified Eagle’s Medium (DMEM; Corning, Manassas, VA, USA). Both RPMI and DMEM mediums were supplemented with 5% fetal bovine serum (FBS; Gibco, New York, NY, USA), and 1% penicillin/streptomycin solution (Sigma-Aldrich, St. Louis, MO, USA). Cells were maintained at 37 °C in a 5% CO_2_ atmosphere and the culture medium was changed every two days. 

### 2.2. Drugs

Cannabinoid Mixture - 8 component solution (CM; Cat. Number = C-219-1ML) was purchased from Sigma-Aldrich (Oakville, ON, Canada) and used at various concentrations (0, 0.1, 1 and 2 μg/mL). 

### 2.3. Cell Viability Assay

Cell proliferation was evaluated using classical MTT assay and confirmed by nucleus staining with Hoechst. As previously described [28,29], 10^5^ cells/well from Ca9-22 and GMSM-K cells were seeded in 12-well plates, cultured overnight and then exposed for 24 h to CM to concentrations (from 0.1 μg/mL to 2 μg/mL ). After cannabinoid exposure, 1/10 dilutions of 5 mg/mL MTT reagent (MTT; Sigma-Aldrich, Oakville, Ontario, Canada) were added, and cells were incubated for 3 h at 37 °C in the dark. Formazan crystals were solubilized using 1 mL of a 0.05 N HCl-isopropanol solution. Next, 4 × 200 µL of lysis buffer was transferred to a 96-well microplate to measure absorbance at 550 nm by an xMark reader (Bio-Rad, Mississauga, ON, Canada). Percentage of proliferation in living cells was determined by using the following formula: % of cell viability = [(OD_550 nm_ (treated cells) − OD (blank))/(OD (control cell) − OD (blank))] × 100. The IC50 of CM was obtained by plotting the percentage inhibition of cell proliferation against the concentration of the cannabinoid mixture. 

### 2.4. LDH Assay

Cell cytotoxicity was evaluated by using LDH assay. As previously described [30], lactate dehydrogenase (LDH) released into the growth media was measured with the LDH Cytotoxicity Assay Kit from BioVision (BioVision, Milpitas, CA, USA). Briefly, 3 × 10^5^ cells per well were seeded in a 6-well plate and left overnight for adhesion, and then, exposed for 24 h to different concentrations of CM. Solutions from the mixture, 50 μL of each supernatant and equivalent volumes from the reconstituted substrate were prepared in triplicates into a 96-well plate before being incubated for 15 min to 20 min at room temperature in the dark until a yellow color developed, and read at 490 nm with an xMark microplate absorbance spectrophotometer (Bio-Rad, Mississauga, ON, Canada). Cells treated with Triton X-100 (1%) were used as a positive control for LDH and the negative one was obtained with untreated. LDH release was calculated with the following formula: % of LDH activity = [cannabinoid mixture (absorbance) − negative control (absorbance)] × 100)/[positive control (absorbance) − negative control (absorbance)]. 

### 2.5. Determination of ROS Levels by Flow Cytometry

Total oxidative stress was evaluated by flow cytometry using ROS markers with the assay protocol from ImmunoChemistry Technologies. Cancer cells were stimulated with CM at 0 μg/mL and 1 μg/mL (IC50) for 24 h. After trypsinization, cells were washed with PBS and were then resuspended in 490 μL RPM1-1640 medium and treated with 10 μL of a ROS Green working solution in the dark for 1 h. The fluorescence intensity of labeled cells was measured with a BD LSR II flow cytometer or a BD FACSCanto II flow cytometer (BD Biosciences). The percentage of positive results was calculated in living cells with FACSDiva Software v. 6.1.3. 

### 2.6. Measurement of Mitochondrial Superoxide

The level of mitochondria-mediated oxidative stress was assessed by using the MitoSOX-Red Mitochondrial Superoxide Indicator (Invitrogen/Thermo Fisher, Rockford, IL, USA). In brief, Ca9-22 cells were treated with the vehicle or with 1 µg/mL of CM for 24 h. Then, cells were washed twice with PBS and incubated for 30 min in the dark with 5 mmol/L of mitochondrial tracking dye (MitoSOX Red; Molecular Probes, Invitrogen), before being analyzed by flow cytometry to determine the percentage of MitoSox-positive cells. 

### 2.7. Mitochondrial Membrane Potential (ΔΨm) Assay

Mitochondrial function, a key indicator of cell health, can be achieved by monitoring changes in membrane potential. Cationic fluorescent dyes are commonly used tools to assess ΔΨm. The latter was evaluated with MitoProbe^TM^ DiOC2(3) Assay Kit from Thermo Fisher according to manufacturer instructions. Briefly, Ca9-22 cells were treated with 1 µg/mL of CM for 24 h. After washing twice with PBS and centrifugation, these were resuspended in 1 mL of PBS at 10^6^ cells/mL and loaded with 5 µL of 10^6^ µM of DiOC_2_(3) for 30 min at 37 °C in the dark. Next, cells were centrifuged and resuspended in 0.5 mL of PBS to perform a flow cytometry analysis using “LSRII” or “CantoII” cytometer instrument from BD Biosciences. 

### 2.8. Intracellular Reduced Glutathione (GSH) Assay

To measure levels of intracellular GSH, we used a kit from ImmunoChemistry Technologies. In brief, ThioBright^TM^ Green reagent was added to the cell suspension (at a dilution of 1:200) and incubated in the dark for 30 minutes at 37 °C. Cells were washed twice with PBS before being harvested by centrifugation and then analyzed with a BD Accuri C6 flow cytometer system (BD Biosciences).

### 2.9. Quantification of Cellular Autophagy

The evaluation of the potential effect of CM on autophagy in oral cancer cells was measured by flow cytometry as previously described [28,29]. Cells were seeded in 60 mm Petri dishes at 10^6^ cells/each and incubated overnight at 37 °C with 5% CO_2_. Then, they were stimulated for 24 h by a vehicle (ethanol) or with 1 µg/mL of CM (this concentration corresponds to the IC50 of cannabinoid mixture). After treatment time, cells were trypsinized, collected and centrifuged, as well as resuspended in 500 μL of a new culture medium containing 1/5 Red staining solution (ImmunoChemistry Technologies, Davis, CA, USA) for 1 h at 37 °C in the dark. Next, cells were washed and suspended in 500 μL fresh 1× assay buffer before acquisition with a BD LSR II system or a BD FACSCanto II system (BD Biosciences) equipped with FACSDiva Software v. 6.1.3 (Mississauga, ON, Canada).

### 2.10. Autophagy Gene Expression by Using RT^2^ Profiler PCR Array

To investigate the differential expression of autophagy-related genes after CM exposure, real-time PCR using RT^2^ Profiler PCR Array Human Autophagy (PAHS-084ZF from Qiagen) was performed to study the effect of CM treatment at 1 µg/mL (IC50 analysis) on Ca9-22 cells according to manufacturer instructions. Briefly, Ca9-22 cells were seeded in a 6-well plate at the density of 3 × 10^5^ cells per well, incubated overnight at 37 °C and 5% CO_2_ and left to adhere. Then, they were treated with 1 µg/mL of CM for 6 h. Total RNA was extracted using a Qiagen RNA extraction kit as directed by the manufacturer. The RNA concentration and purity have been measured with the NanoDrop spectrophotometer (Thermo Fisher). RNA was reverse-transcribed into cDNAs with the RT2 First Strand Kit (Qiagen, NV). PCR array was performed using PCR mixture components, containing 1350 µL SYBR Green Mastermix, 102 µL cDNA and 1248 µL RNase-free water. Next, 25 µL of the mixture was added to each well of the RT^2^ profiler plate. This was centrifuged for 1 min at 1000× *g* at 25 °C and the real-time PCR was carried out. The array displays the expression of 84 key genes, which are modulated in apoptosis and autophagy. Finally, the data were analyzed using the 2^-ΔΔCT^ method for relative gene expression as well as the fold changes between non-exposed and CM-treated cells being evaluated. A gene was considered up- or down-regulated when the fold change is more than twice the original number. CT values were derived from an Excel file to make a table with them, which is then uploaded on the web portal for data analysis at http://www.qiagen.com/geneglobe, accessed on 1 September 2022. Control samples were untreated cells; however, test groups were cells treated with 1 µg/mL of CM. The gene expressions we obtained were normalized depending on automatic selection from a full panel of reference genes. The experiments were performed three separate times.

### 2.11. Cell Apoptosis and Programmed Cell Death by Flow Cytometry with Annexin V Assay

Ca9-22 cells were seeded in a 60 mm cell culture/Petri dish at 10^6^ cells/each. At 80% confluence, cells were stimulated with different concentrations of CM for 24 h at 37 °C and 5% CO_2_. After this step, cells were trypsinized, counted and incubated with 5 µL Annexin V-FITC and 10 µL propidium iodide (BD Biosciences, Mississauga, ON, Canada) for 30 min in the dark at room temperature. Cells were analyzed using either BD LSR II or BD FACSCanto II cytometer (BD Biosciences) equipped with FACSDiva Software v. 6.1.3. The experiment was repeated three times. In addition, caspase activity was also assessed with the Caspase Detection Kit (FITC-VAD-FMK) from Millipore Corp. (Burlington, MA, USA). 

### 2.12. Cell Morphology

Oral cancer cells (Ca9-22) were seeded on sterile glass coverslips at 10^5^ cells/slide immersed in RPMI-1640 stimulated for 24 h with various concentrations of CM (0.1 μg/mL and 2 μg/mL). After, cells were washed with PBS and slides were observed under an epifluorescence microscope (Nikon Optiphot) and photographed with a digital camera (Nikon COOLPIX 995).

### 2.13. DNA Damage Assessment by Flow Cytometry

To evaluate the effect of 1 µg/mL of CM on oral cancer cells, analysis of γH2A.X expression by flow cytometry was performed as previously described [28,29,30]. In brief, after treating Ca9-22 cells with CM for 24 h, they were trypsinized and then fixed in 75% ethanol for 20 min. Afterward, a permeabilization solution containing 1% BSA/0.2% Triton/1× PBS was added to the cells and they were next incubated in the dark at 4°C overnight with the phospho-histone γH2A.X (Ser139) monoclonal antibody from Santa Cruz Biotechnology (Santa Cruz, CA, USA) at a dilution of 1/100. Cells were then washed twice with PBS and incubated for 1 h with an Alexa Fluor 488-conjugated secondary antibody from Santa Cruz Biotechnology at a 1:100 ratio. Acquisition and analysis were performed with a BD flow cytometry system (BD FACSCanto II).

### 2.14. Cell Migration Determined by Wound-Healing Assay

As it was described in Semlali et al. [28,29], Ca9-22 cells were seeded in a 6-well plate and cultured until they reach 100% confluence. Cell monolayers were subjected to a scratch in the shape of a cross with a sterile pipette tip. Immediately after, cells were stimulated with different concentrations of CM (0.1 µg/mL to 2 µg/mL) and incubated at 37 °C for 6 h before taking a photograph of each well with an inverted microscope. The cell migration was analyzed by image-processing software that was able to measure the distance between opposite edges of the scratch at each time point. All wells were then compared, based on their percentage of closure. 

### 2.15. Flow Cytometric Analysis for Studies on Cancer Pathways

As described in an earlier study we conducted [31], the phosphorylation levels of NF-κB STAT1, STAT3, STAT5, p38 and ERK1/2 were determined by flow cytometry as previously reported [32]. Briefly, cells were washed with DPBS + 2% FBS and then fixed in formaldehyde (1.5%) for 20 min at room temperature (RT). After this step, cell washing was performed once again and they were incubated afterward for 20 min in methanol/DPBS (90% v/v), on ice, to permeabilize them. An additional wash was performed, and cells were labeled, for 30 min, with 5 μL of Alexa Fluor 647 mouse anti-human pSTAT1 (clone K51-856), pSTAT3 (clone 49/p-stat3), pSTAT5 (clone 47/Stat5[pY694]), pERK1/2 (clone 20A), phosphorylated p38 (clone 36/389pT180/pY182), all from BD Biosciences and NF-κB p65 (Clone B33B4WP; Thermo Fisher). After a final wash, cells were suspended in 300 μL of DPBS before being analyzed by flow cytometry (BD Accuri C6, BD Biosciences).

### 2.16. Statistical Analysis

The significant difference between experimental (treated) groups and controls (untreated) was evaluated by Student’s *t*-test in GraphPad Prism 7 Software. Data are represented as mean ± SEM. ^*^*p*-value < 0.05 was considered as statistically different. 

## 3. Results

### 3.1. CM Selectively Inhibits Oral Cancer Cell Proliferation and Promote Their Cytotoxicity

To assess the effect of different concentrations of CM on the growth of GMSM-K gingival normal cells and Ca9-22 oral cancer cells, an MTT assay was performed. Results revealed that a dose of CM ≥ 1 μg/mL significantly inhibited Ca9-22 cells from growing at 24 h by around 50% (IC50~1 μg/mL) whereas no significant inhibition in non-tumor-derived epithelial cells (GMSM-K) was observed (Figure 1A). In addition to the MTT assay, an LDH assay was employed to evaluate the cell cytotoxicity and results showed that CM had a statistically significant dose-dependent growth inhibitory effect only on Ca9-22 but not on GMSM-K (Figure 1B) and thus support its selective effect on oral cancer cells. not shown). 

### 3.2. CM Exposure Results in Both Total Intracellular ROS and Mitochondrial ROS Induction and Provokes Mitochondrial Membrane Potential Inhibition

Many natural products can strongly enhance intracellular ROS levels, disrupt intracellular redox homeostasis, and eventually cause tumor cell apoptosis and autophagy [33]. Therefore, we investigated whether CM-induced cytotoxicity in oral cancer cells was associated with a modulation of ROS levels. Thus, treatment of Ca9-22 cells with 1 µg/mL of CM for 24 h showed a significantly increased intracellular ROS (from 43.1% ± 4.1% to 69.7% ± 9.3%; Figure 2A). Moreover, we looked at the ROS production inside mitochondria using the probe MitoSOX and found that CM potently enhanced the MitoSOX generation in Ca9-22 cells. Indeed, MitoSOX went from 9.8% ± 0.8% in untreated cells to 56.25% ± 12.2% in CM-treated Ca9-22 cells (Figure 2B). These results together indicate that CM at low concentration induces accumulation of mitochondrial stress oxidative in oral cancer cells. Furthermore, alterations in the mitochondrial membrane potential (ΔΨm) dropped significantly, ranging from 94.75% ± 1.9% in test groups to 52.25% ± 8.68% in controls (Figure 2C). In addition to ROS, intracellular GSH, which plays a pivotal role in retaining homeostasis and helping cells to escape apoptosis was also increased following exposure to CM (from 62.25% ± 8.83% in untreated cells to 99.29% ± 0.63% in CM-treated cells; Figure 2D). 

### 3.3. CM Stimulates Autophagy in Oral Cancer Cells

Generated mitochondrial ROS has been associated with autophagy [7,34] which prompted us to investigate a potential autophagic effect of CM treatment on Ca9-22 cells. Our results showed a slight but significant increase in the percentage of autophagy, ranging from 61.25% ± 1.4% to 75.55% ± 1.9%, respectively, for untreated and CM-treated Ca9-22 cells (Figure 3).

In addition, using QIAGEN RT^2^ Profiler PCR Array Human Autophagy, we analyzed 84 genes that are associated with autophagic response. This analysis revealed significant modulation of several genes’ expression between CM-treated cells and untreated cells (Figure 4A). Among them, 46 genes involved in different steps of autophagy were modulated more than twice, and 14 genes were considered components of the autophagy machinery, including: AMBRA1 (53.08 fold), ATG16L1 (10.3 fold), ATG16L2 (25.99 fold), ATG4A ((−2.69 fold), ATG4C (38.72 fold), ATG4D (36.50 fold), ATG9B (10.45 fold), IRGM (6.04 fold), GABARAPL1 (4.29 fold), LAMP1 (−4.26 fold), MAP1LC3A (3.18 fold), MAP1LC3B (3.39 fold), ULK1 (−1221.98 fold), WIPI1 (277.24 fold) and two genes implicated in the ubiquitination process: ATG3 (134.83 fold) and HDAC6 (−2.07 fold). Additionally, from the 47 genes identified, 30 genes were involved in the regulation of the autophagy process such as: (AKT1 (−2.58 fold), APP (−4.72 fold), BID (−3.75 fold), CDKN2A (−2.61 fold), CLN3 (−2.09 fold), CTSB (−3.36 fold), CTSD (−2.35 fold), CTSS (4.41 fold), GAA (2.09 fold), CXCR4 (3.32 fold), DAPK1 (3.07 fold), EIF4G1 (3.29 fold), FADD (−3.66 fold), HDAC1 (−3.11 fold), HGS (91.46 fold), IFNG (12.55 fold), IGF1 (3.18 fold), INS (10.67 fold), MAPK8 (1205.15 fold), MTOR (−2.65 fold), PIK3CG (3.53 fold), PIK3R4 (−2.61 fold), RB1 (−2.39 fold), RPS6KB1 (1031.12 fold), SQSTM1 (4.08 fold), TGFB1 (−2.01 fold), TMEM74 (−2.14 fold), TNF (5.72 fold), TNFSF10 (−2.52 fold), ULK2 (2.72 fold)) (Figure 4B ).

### 3.4. CM Induces Apoptosis in Oral Cancer Cells Via Caspase Activation

Both ROS and autophagy have been reported to play an important role in cellular apoptosis or programmed cell death (Ferrari, n.d.) (Ferrari, 2000; Jung et al. Experimental & Molecular Medicine (2020) 52:921–930) and we found that treatment of Ca9-22 cells with various concentrations of CM (0 µg/mL and 1 µg/mL) for 24 h resulted in a dramatic increase of apoptotic cell death in a dose-dependent manner, as shown in Figure 5A. Indeed, the percentage of apoptosis went from 8.1% in untreated cells to 38% when they were exposed to 1 µg/mL of CM. Blocking oxidative stress with N-acetylcysteine (NAC) or autophagy with the inhibitor 3-methyladenine (3-MA) completely suppressed apoptosis induced by CM (Figure 5A). Our data suggest that CM induces apoptosis in oral cancer cells via ROS-based and autophagy processes. Furthermore, during apoptosis, caspases are activated to ensure that programmed cell death occurs with less damage to surrounding tissues and cell components degrade in a well-controlled manner [35]. Thus, we looked at caspases and we found that CM-induced apoptosis was associated with an increased expression of them. Indeed, the percentage of caspase-positive cells went from 25.05% ± 0.3% in controls to 51.9% ± 8.7% in test groups, suggesting that CM at a low concentration significantly enhanced caspase activity in Ca9-22 cells (Figure 5B).

### 3.5. CM Affects Ca9-22 Cell Morphology and Induces DNA Damage

Apoptosis and DNA damage are key features of excessive oxidative stress (Li et al., 2006; Peng et al., 2021); thus, we looked at the effect of CM on cell morphology and DNA damage. Our results showed that CM caused dose- and time-dependent changes in the morphology of Ca9-22 cells with remarkably longer neurites having ceased to grow/divide (Figure 6A). In parallel with morphology assessment, we dug into the expression of the γH2AX DNA damage marker in Ca9-22 cells through exposure to 1 µg/mL of CM. Our results revealed significantly increased expression of γH2AX-positive cells following treatment (86.6% ± 2.6%) compared to none (32.85% ± 4.15%; Figure 6B).

### 3.6. Anti-Migratory Effect of Cannabinoid Mixture in Ca9-22 Cells

One fundamental feature of cancer is the ability of tumor cells to migrate from their original location to other parts of the body. Therefore, the anti-migratory effect of CM was evaluated in Ca9-22 cells. These were scratched and treated with 1 µg/mL of CM for 6 h, and the degree of Ca9-22 cell migration was examined. Compared to the controls, CM markedly suppressed the migration of Ca9-22 cells (Figure 7) supporting its anti-cancer potential.

In addition, numerous pathways including MAPKase have been shown to play an important role in cancer cell survival. Thus, we looked at the expression of pNF-κB (76.55 ± 5.8 vs. 49.65 ± 2.9% of positive cells), pSTAT1 (18.35 ± 8.9 vs. 7.63± 3% of positive cells), pSTAT3 (17.78 ± 6.7 vs. 2.75 ± 0.3% of positive cells), pSTAT5 (65.16 ± 11.4 vs. 51.43 ± 5.3% of positive cells), p38 (53.84 ± 1.9 vs. 33.7 ± 12.6% of positive cells) and pERK1/2 (60.29 ± 13.07 vs. 50.9 ± 4.9% of positive cells) in both and CM-treated and untreated Ca9-22 cells as shown in Figure 8. We found that CM treatment resulted in a significantly down-regulated expression of the mentioned signaling molecules, thus supporting its cancer-fighting potential.

## 4. Discussion

We recently reported the ability of cannabis derivatives such as Δ^9^-THC and Δ^8^-THC to kill oral cancer cells by disrupting their cell cycle and increasing apoptosis, autophagy, oxidative stress and DNA damage [28]. However, to the best of our knowledge, the effect of a mixture of cannabinoids on oral cancer cell cytotoxicity and its related mechanisms have not been investigated so far. Herein, we show that treatment of oral cancer cells with a low concentration (between 0.1 µg/mL and 1 µg/mL) of this mixture resulted in their selective cell death and consequently highlight the anti-cancer potential of cannabinoid derivatives.

Oral carcinogenesis is believed to be a multistep process derived from an accumulation of cellular alterations induced by carcinogens. Fundamental to this process are those genes regulating cell division, cell cycle progression and DNA replication and repairs. This information led us to assess the effect of CM on GMSM-K gingival normal cells and Ca9-22 oral cancer cells in terms of growth and cytotoxicity. Our results revealed a selective inhibition of Ca9-22 cell proliferation associated with low cytotoxicity (Figure 1, panels A and B), suggesting a potential anti-cancer effect of CM. This antiproliferative activity of CM prompted us to look at oxidative stress in intracellular and mitochondrial compartments. Our results indicated a significantly increased level of ROS induced by CM treatment in both compartments (Figure 2, panels A and B). This outcome was in accordance with a report made by Park et al. who showed that the endocannabinoid anandamide (AEA) had an anti-cancer effect on head and neck squamous cell carcinoma (HNSCC) cell lines, which was mediated by increased ROS levels [36]. In addition, this increase in mitochondria was associated with a significant decrease in their ΔΨm suggesting an impaired functional capacity of the organelle (Figure 2, panel C). This finding further supports the anti-cancer ability of CM as a marked decrease in mtMP featured slow proliferation and strong tumorigenicity impairment [37]. Moreover, loss of mitochondrial membrane potential has been linked to ROS-mediated apoptosis [38,39] and we found the latter process had a significant increase in CM-treated oral cancer cells. Additionally, GSH, which plays an important role to prevent cell damage induced by oxidative stress, was significantly increased but probably not enough [40,41]. Autophagy, a complex evolutionary conserved catabolic process in which cells self-digest intracellular organelles in order to regulate their normal turnover and remove the damaged ones with compromised function, to further maintain homeostasis has been associated with mitochondrial-generated ROS [7,34]. As expected, our data revealed a significant up-regulation of autophagy. Indeed, several genes were modulated by CM including those considered to be involved in autophagy machinery components, ubiquitination and the regulation of this process. Furthermore, we identified many genes that possess a dual role in autophagy and apoptosis, such as death-associated protein kinase 1 (DAPK1) which was up-regulated 2.03 times with CM treatment. The latter is part of the serine/threonine (Ser/Thr) family, with tumor suppressive function known to phosphorylate Beclin 1 [42,43]. Bcl-2 family genes were also down-regulated by CM treatment and thus could promote overexpression of Beclin 1-dependent autophagy in accordance with the work of Sun et al., in which they demonstrated that this can lead to an increased anti-cancer drug-induced apoptosis in cancer cells [44]. As both ROS and autophagy play an important role in cell death, we looked at apoptosis in Ca9-22 cells following exposure to CM and we found that it was significantly increasing, which was completely abrogated in the presence of NAC (ROS inhibitor) or 3-MA (autophagy inhibitor). Furthermore, this apoptosis was mediated through numerous pathways including increased caspase expression.

Along with apoptosis, DNA damage is one of the key features of excessive oxidative stress [45,46], it was significantly increased as assessed by the expression of histone γH2AX that was associated with a drastic morphological change in Ca9-22 cells and thus could explain the CM-induced apoptosis reported herein. In addition to all mentioned effects associated with CM and described, our results highlighted the ability of CM to inhibit Ca9-22 cell migration. This finding was in accordance with the reduced activation of important signaling pathways, such as NF-κB and MAPKase, which govern tumor growth and cancer cell invasion [47,48,49], and thus further support a potent anti-cancer effect for CM. 

In summary, our study demonstrates that a mixture of cannabinoids particularly at a dose of 1 µg/mL was able to inhibit oral cancer cell proliferation through diverse mechanisms including apoptosis and autophagy. 

## 5. Conclusions

Our findings provide evidence that cannabinoids at limited concentrations could serve as an effective therapeutic strategy to target human oral cancers.

## Figures and Tables

**Figure 1 cancers-14-04924-f001:**
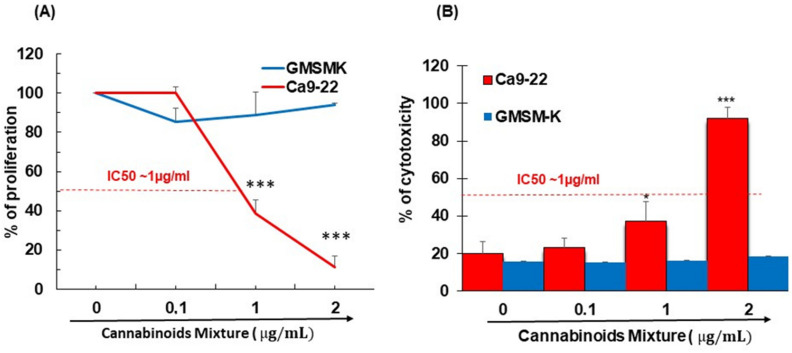
Proliferation of cancer cell line (Ca9-22) and non-cancerous cell line (GMSM-K) and after treatment with increasing concentrations of CM. (**A**) The cell growth was detected by the MTT assay after stimulation with various concentrations of CM for 24 h. (**B**) Cell cytotoxicity was determined after CM exposure for one day. It was measured by LDH assay. The results for each cell line were normalized to untreated controls (*n* = 6 biological replicates, data are expressed as mean ± SEM).* *p* < 0.05, *** *p* < 0.0005.

**Figure 2 cancers-14-04924-f002:**
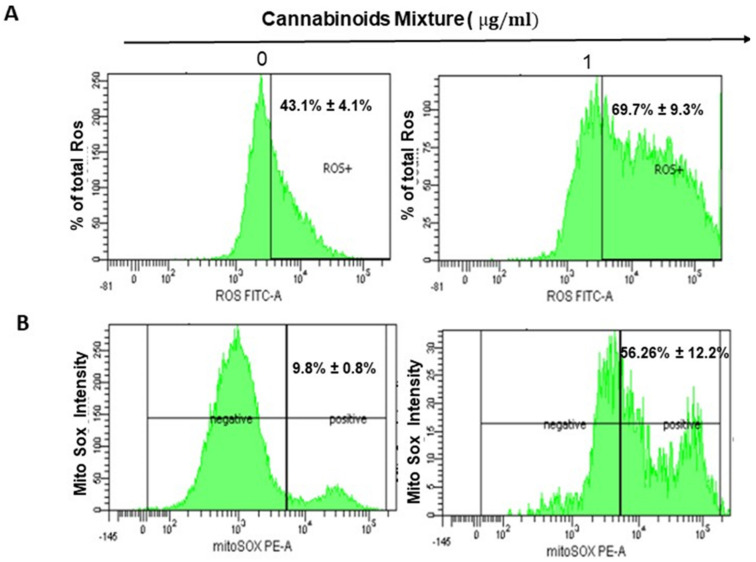
CM promotes oxidative stress and autophagy. (**A**) CM induces ROS expression in oral cancer cells. ROS levels were measured by flow cytometry using markers in the protocol (*n* = 4). (**B**) Measurement of mitochondrial superoxide. Generation of mitochondrial ROS, in CM-treated and untreated Ca9-22 cells, was also assessed by this technique with MitoSOX staining, highly selective to detect superoxide in mitochondria from living cells. (**C**) Mitochondrial membrane potential was evaluated by flux cytometry using a MitoProbe^TM^ DiOC2(3) Assay Kit from Thermo Fisher. Ca9-22 cells were treated with 1 µg/mL of CM for 24 h. Next, 10^6^ cells were loaded with 5 µL of 10^6^ µM DiOC_2_(3) for 30 min at 37°C in the dark, before performing a flow cytometry analysis (*n* = 4). (**D**) Cells were exposed to CM at 1 µg/mL for 24 h and subjected to GSH staining, then analyzed by flow cytometry. Results are expressed as mean ± SEM (*n* = 4).

**Figure 3 cancers-14-04924-f003:**
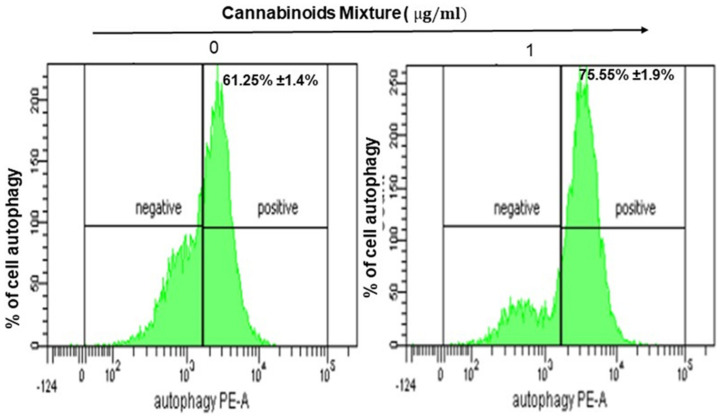
CM induces autophagy: Representative flow cytometry histogram of Ca9-22 cells stained with the autophagy probe. Results are expressed as mean ± SEM (*n* = 4).

**Figure 4 cancers-14-04924-f004:**
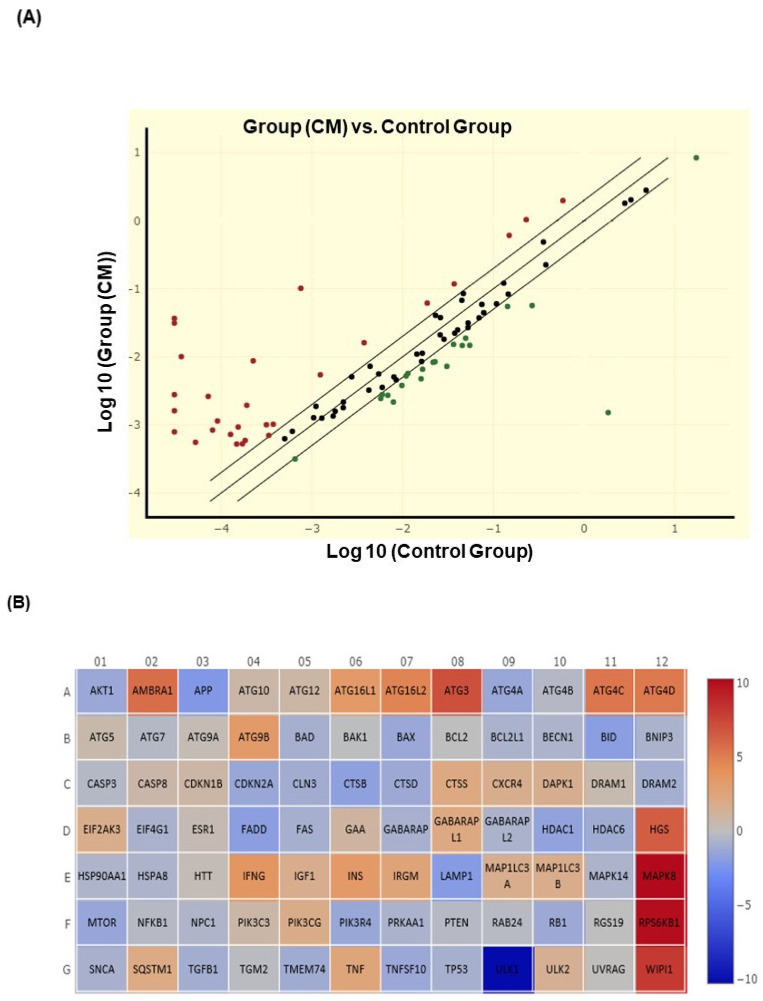
CM modulates genes involved in the regulation of the autophagy process. (**A**) Differential expression of autophagy-related genes between untreated and CM-treated Ca9-22 cells using RT^2^ Profiler PCR Array tests. A total of 46 differential expressed autophagy genes were identified (up- or down-regulated more than twice). (**B**) Visualization of log2 fold changes and up/down-regulation by cannabinoid mixture compared to untreated cells.

**Figure 5 cancers-14-04924-f005:**
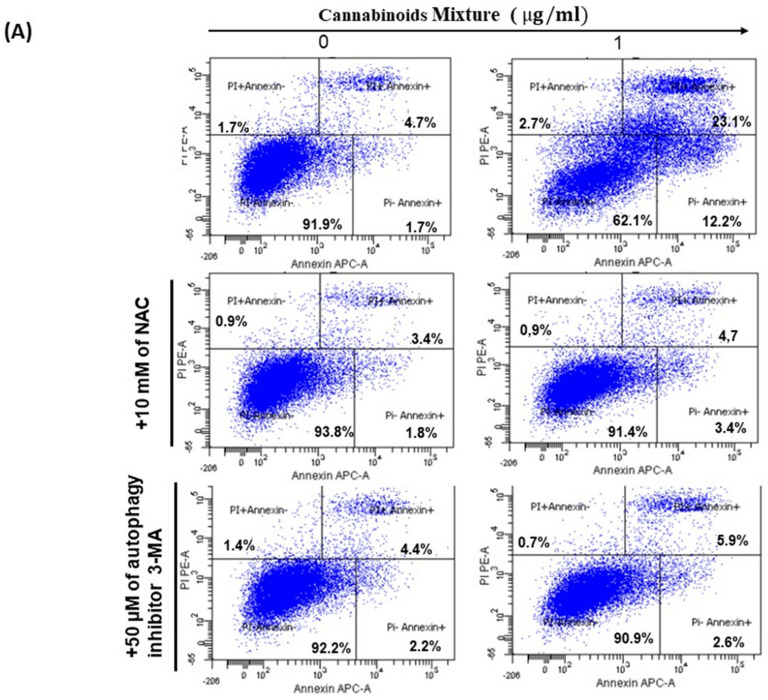
CM promotes apoptosis in Ca9-22 cells. (**A**) Flow cytometry assay using Annexin V and PI to measure apoptotic cell death in oral cancer cells after treatment with 1 μg/mL of CM (*n* = 3), without or with 10 mM of NAC or 50 µM of 3-MA. (**B**) Activation of caspases by CM in Ca9-22 cells detected by flow cytometry.

**Figure 6 cancers-14-04924-f006:**
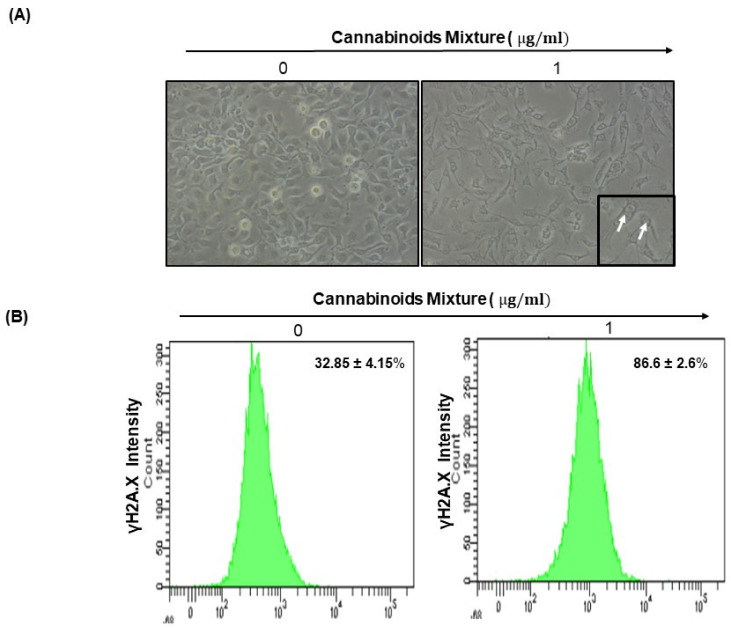
Cannabinoid mixture affects Ca9-22 cell morphology and induces DNA damage. (**A**) Morphology of Ca9-22 cells that were seeded at low density in the presence of CM for 24 h as described in Materials and Methods. (**B**) Flow cytometry expression profiling of γH2A.X after CM treatment. Cells were grown on Petri dishes and stimulated by CM at 1 µg/mL (IC50 concentration) (*n* = 3). The white arrow represents the nucleus damage caused by CM.

**Figure 7 cancers-14-04924-f007:**
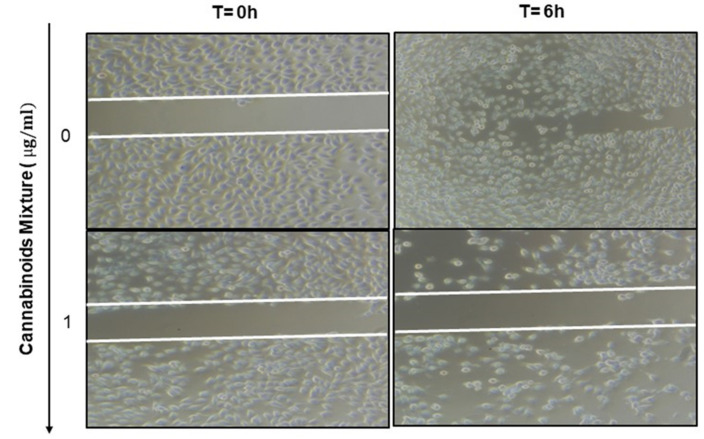
Wound-healing assay and transcription factor expression. The effect of CM on cell migration was determined by the scratch method. Unstimulated Ca9-22 cells were compared to CM-treated Ca9-22 cells. The cell migration was analyzed by image-processing software that was able to measure the distance between opposite edges of the scratch at each time point. All wells were then compared, based on their percentage of closure.

**Figure 8 cancers-14-04924-f008:**
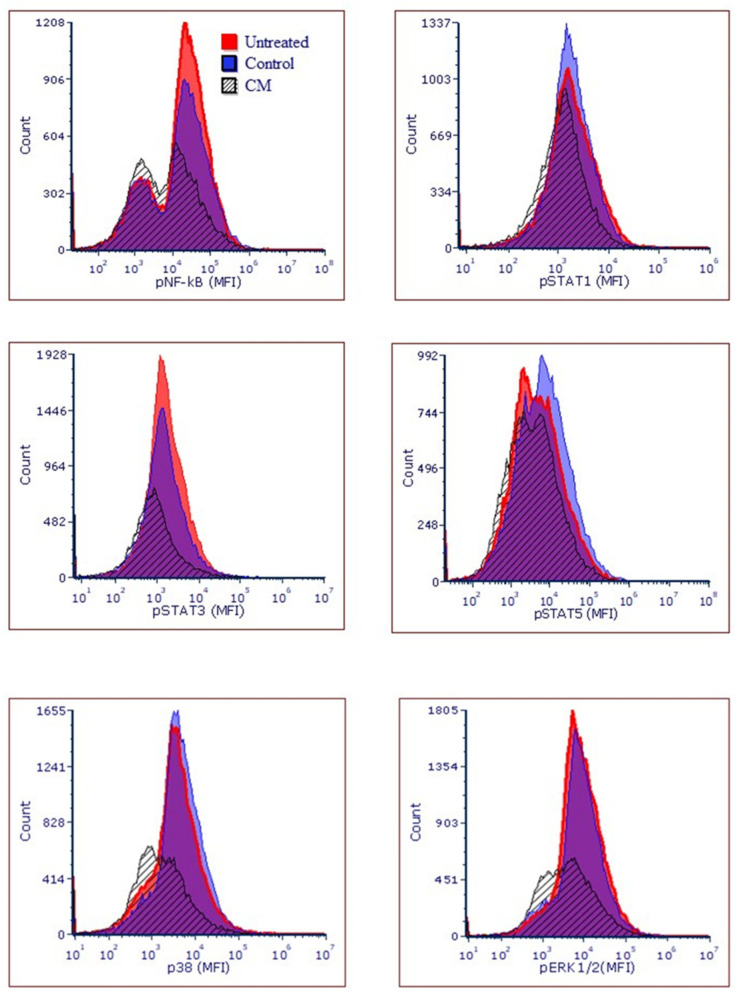
The expression of pNF-κB (76.55 ± 5.8 vs. 49.65 ± 2.9% of positive cells), pSTAT1 (18.35 ± 8.9 vs. 7.63 ± 3% of positive cells), pSTAT3 (17.78 ± 6.7 vs. 2.75 ± 0.3% of positive cells), pSTAT5 (65.16 ± 11.4 vs. 51.43 ± 5.3% of positive cells), p38 (53.84 ± 1.9 vs. 33.7 ± 12.6% of positive cells) and pERK1/2 (60.29 ± 13.07 vs. 50.9 ± 4.9% of positive cells) in both and CM-treated and untreated Ca9-22 cells.

## Data Availability

Not applicable.

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
