# Peer review of "Oxidative Stress and Autophagy Mediate Anti-Cancer Properties of Cannabis Derivatives in Human Oral Cancer Cells"

_cancers, 2022, doi:10.3390/cancers14194924_

Round 1

Reviewer 1 Report

Dear Authors:

The manuscript "Oxidative stress and autophagy mediate anti cancer proprieties of cannabis derivatives in human oral cancer cells." by Loubaki et al has demonstrated that cannabinoids could potentially have a beneficial effect for oral cancer therapy. I have just a few suggestions.

1. The manuscript needs linguistic improvement.

2. Some references or information are missing. 

In introduction, Page 2, Line 54-62:", the involvement of reactive oxygen species (ROS) in cancer development has been studied for decades and there is sufficient evidence that implicates them in the multistage theory of carcinogenesis as they are proposed to cause diverse DNA alterations like: punctual mutations, DNA base oxidation, strand breaks, mutation of tumor suppressor genes and can induce overexpression of proto-oncogenes [8, 9]. Moreover, increased ROS level has been linked to tissue injury and/or damage to intracellular components and participate in a wide range of crucial physiological processes such as cell cycle progression, antiapoptotic mechanisms, invasion, metastasis and angiogenesis, which contribute to different types of cancer pathogenesis [10]." There are more reviews demonstrated the roles of mitochondria and ROS in cancer development. (please cite: 1. Advances in the Prevention and Treatment of Obesity-Driven Effects in Breast Cancers. Front Oncol. 2022 Jun 22;12:820968. doi: 10.3389/fonc.2022.820968. PMID: 35814391; PMCID: PMC9258420.

2. An Epigenetic Role of Mitochondria in Cancer. Cells. 2022 Aug 13;11(16):2518. doi: 10.3390/cells11162518. PMID: 36010594; PMCID: PMC9406960.

3. Mitochondrial mutations and mitoepigenetics: Focus on regulation of oxidative stress-induced responses in breast cancers. Semin Cancer Biol. 2022 Aug;83:556-569. doi: 10.1016/j.semcancer.2020.09.012. Epub 2020 Oct 6. Erratum in: Semin Cancer Biol. 2022 Jul 16;: PMID: 33035656.)

Best,

Reviewer 2 Report

Dear Authors,
This paper addresses a very interesting topic, with well organized structure and with excellent results,
however, I would recommend some modifications before considering its publication. Below these are some
suggestions for You:
1. Introduction:
a. line 51: in 2022?
b. I suggest mentioning about etiological factors (in the Introduction or Discussion), like
smoking or alcohol drinking, dysbiotic flora (4.5 subsection:
https://doi.org/10.3390/cancers13174439), or HPV
(https://doi.org/10.5114/ada.2021.107269),
2. Material and methods:
a. All methods adequately described
3. Results:
a. Figure 4 - barely visible, figure 4C should be detached, and 4B larger, resolution should be
better
4. Discussion:
a. I suggest mentioning about etiological factors (in the Introduction or Discussion)
5. Conclusions:
a. I suggest to create Conclusions section, according to the Instruction for Authors, for lines
462-466 from the Disucssion
6. References:
a. The article is well documented with a vast number of references.
7. Layout and editorial
a. line 95 - mind the gap between ‘in’ and ‘RPMI-’
b. line 327 - double coma
c. line 434 - mind the gap between ‘stress’ and ‘was’
d. line 435 - mind the gap between ‘cells.’ and ‘Autophagy’
e. line 461 - lacking space before ‘and thus’
Best regards and good luck

Reviewer 3 Report

In this study, the authors show that cannabis could induce oral cell death by inducing DNA damage and activating the mechanisms of autophagy and apoptosis, This is a potentially interesting study. However, there are several major points needed to be addressed: 

1,In figure 1A, why the CM could inhibit GMSM-k cell proliferation in 0.1 ug/ml? and why did >15% of cells cytotoxicity without treatment in Figure 1B.

2,The resolution of picture 2 is not high and needs to be replaced.

3,What does the long straight line in Figure C represent?

4,Please add scale bar in figure 6a. What is marked by the white arrow?

5,In figure 7A, the cells in the top right are larger than the cells in left, why?

6,In Methods and Materials, why did the authors use 5% FBS to culture cells? It is usually 10%. In line 128, what is the 3.105 cells stand for?

Round 2

Reviewer 1 Report

strongly suggestion for publication.

Author Response

Thank you for your times and your nices comments

All the best

Reviewer 2 Report

Dear Authors,

  1. This is a revised version of your original manuscript. Unfortunately it has not been corrected according to my suggestions.  Please revise it again. 

    1. Firstly, not all suggested papers were used

    2. Besides, as you wrote, an etiology section was included, indeed - you did not include it in the manuscript (I hope it’s just a mistake)

    3. You did not reply to the suggestion concerning Conclusions

    4. You did not provide any changes in Figure 4

Best regards and good luck

Author Response

Thanks your for your suggestion

Reviewer 3 Report

It would be nice to see the authors' revisions to the manuscript, but I don't see the authors' revisions to Figure 7A, the images are at different magnifications. Also, I suggest that the authors redraw Figure 4C, which looks too incomprehensible.

Author Response

Thank you for your nice suggestion

Round 3

Reviewer 2 Report

Dear Authors,

This is a revised version of your original manuscript. You have corrected it according to my suggestions. Below these are some minor suggestions for you

  • The references list is doubled - remove unnecessary one 

  • Conclusions should be the next section , not the part of the discussion

Best regards and good luck

Author Response

Thank you for your support

All it done

All the best

Reviewer 3 Report

Agree to publish

Author Response

Thank you for your nice support

All the best